# Exploring the hidden interior of the Earth with directional neutrino measurements

Michael Leyton[1,2,3], Stephen Dye[4] & Jocelyn Monroe[2,3,5]

Roughly 40% of the Earth's total heat flow is powered by radioactive decays in the crust and mantle. Geo-neutrinos produced by these decays provide important clues about the origin, formation and thermal evolution of our planet, as well as the composition of its interior. Previous measurements of geo-neutrinos have all relied on the detection of inverse beta decay reactions, which are insensitive to the contribution from potassium and do not provide model-independent information about the spatial distribution of geo-neutrino sources within the Earth. Here we present a method for measuring previously unresolved components of Earth's radiogenic heating using neutrino-electron elastic scattering and low-background, direction-sensitive tracking detectors. We calculate the exposures needed to probe various contributions to the total geo-neutrino flux, specifically those associated to potassium, the mantle and the core. The measurements proposed here chart a course for pioneering exploration of the veiled inner workings of the Earth.

[1] Institut de Física d'Altes Energies, Barcelona Institute of Science and Technology, Facultat Ciencies Nord, Campus UAB, Bellaterra 08193, Spain. [2] Department of Physics, Royal Holloway University of London, Egham Hill, Egham TW20 0EX, UK. [3] Department of Physics and Laboratory for Nuclear Science, Massachusetts Institute of Technology, 77 Massachusetts Avenue, Cambridge, Massachusetts 02139, USA. [4] Department of Physics and Astronomy, University of Hawaii, 2505 Correa Rd., Honolulu, Hawaii 96822, USA. [5] High Energy Accelerator Research Organization (KEK), 1-1 Oho, Tsukuba, Ibaraki 305-0801, Japan. Correspondence and requests for materials should be addressed to M.L. (email: leyton@cern.ch).

The Earth's surface heat flow[1] of $47 \pm 2$ TW is fuelled in part by the radioactive decay of uranium (U), thorium (Th) and potassium (K) in the crust and mantle. The antineutrinos produced by these decays, called geo-neutrinos (geo-$\bar{v}_e$s) due to their geophysical origin, give us important clues about the composition of the Earth's interior and the size and sources of its internal heating due to radioactivity, both in the current epoch and throughout its evolution. Geo-$\bar{v}_e$s produced by decays from the radioactive cascades of $^{238}$U and $^{232}$Th have previously been measured by the KamLAND[2–4] and Borexino[5–7] experiments via the detection of inverse beta decay on free protons, $\bar{v}_e p \rightarrow e^+ n$, where $\bar{v}_e$, $p$, $e^+$ and $n$ denote an electron antineutrino, proton, positron and neutron, respectively.

While previous geo-$\bar{v}_e$ measurements[3] have been used to constrain the portion of the Earth's surface heat flow due to radioactivity[8] ($38 \pm 23\%$), they are insensitive to the contributions from $^{40}$K and $^{235}$U, whose geo-$\bar{v}_e$ spectra fall entirely below the 1.8 MeV threshold of inverse beta decay on free protons. Specifically, the $^{40}$K geo-$\bar{v}_e$ flux has never been measured, although $^{40}$K decays may have been the dominant heat source for the primordial Earth, due to the shorter half-life of $^{40}$K relative to $^{238}$U and $^{232}$Th. Measuring the $^{40}$K geo-$\bar{v}_e$ flux would provide a unique view to the Earth's origin and formation, as well as the composition of its interior. In order to gain sensitivity to interactions of $^{40}$K geo-$\bar{v}_e$s, a reaction with an energy threshold lower than the 1.3 MeV endpoint of the $^{40}$K beta ($\beta^-$) decay spectrum is required.

Moreover, existing measurements of geo-$\bar{v}_e$s from $^{238}$U and $^{232}$Th are non-directional and therefore do not provide model-independent information about the location or spatial distribution of geo-$\bar{v}_e$ sources in the crust and mantle. As a consequence, resolution of crust and mantle contributions depends on geochemical modelling, with the addition of associated uncertainties. Similarly, while the Earth's surface heat flow ($47 \pm 2$ TW) is well measured[1], the contributions from radiogenic heating[3,8], secular cooling of the mantle and heat flow out of the core[9] are relatively poorly constrained. Tightening these constraints would provide better insight to the thermal evolution of our planet.

Direction-sensitive measurements of geo-$\bar{v}_e$s could additionally help unravel a long-standing mystery surrounding the source of heat needed to power the geo-magnetic field, vital for protecting life on Earth from the Sun's radiation. In particular, constraints on the presence and concentration of radioactive elements in the Earth's core[10,11] are critical for understanding the geo-dynamo and the evolution of the inner core. Recent measurements[12] at high pressure (1.5 GPa) and high temperature ($\geqslant 1,400\,^\circ$C) indicate that a sulfur-rich core would partition uranium strongly and thorium slightly, resulting in U, Th concentrations of up to 10 p.p.b. in the core and producing 2.4 TW of heat, sufficient to power the geo-dynamo, even without a contribution from $^{40}$K decay[10]. Direction-sensitive measurements of geo-$\bar{v}_e$s could shed light on this mystery by disentangling individual contributions from the crust, mantle and core to the total geo-$\bar{v}_e$ flux.

We propose a method for measuring the geo-$\bar{v}_e$ flux from $^{40}$K, the Earth's mantle, and the Earth's core using neutrino-electron ($v_\ell e^-$) elastic scattering in low-background, direction-sensitive tracking detectors. Unlike inverse beta decay, the elastic scattering reaction discussed here, $v_\ell e^- \rightarrow v_\ell e^-$, has no energy threshold and is therefore sensitive to the $^{40}$K geo-$\bar{v}_e$ flux. Furthermore, elastic scattering preserves directional information, since the direction of the outgoing electron is well correlated with the direction of the incoming neutrino, which can be exploited in order to reject backgrounds and disentangle the various sources of geo-$\bar{v}_e$s. While backgrounds from radioactivity can be quantified, controlled and minimized, the dominant source of irreducible background to $\bar{v}_\ell e^-$ elastic scattering comes from

elastic scattering of other neutrinos, primarily solar $v_e$s. However, a detector with the ability to measure the direction of incident neutrinos has additional rejection power against such events since the solar-$v_e$ background points back to the Sun, unlike much of the geo-$\bar{v}_e$ signal. Using the electron angle, energy and the position of the Sun at the time of interaction, we show that a direction-sensitive detector can, in principle, discriminate between the solar-$v_e$ background and the geo-$\bar{v}_e$ signal.

In the following, we estimate the sensitivity of a direction-sensitive detector to the flux of geo-$\bar{v}_e$s from $^{40}$K decays, the Earth's mantle and the Earth's core. Detector resolution, efficiency and energy thresholds are included in the analysis, based on the performance of previous detectors[13,14]. We construct detailed maps of the directions to geophysical structures[15,16] for three underground sites where large-scale neutrino observatories are currently operating and where such a detector could be deployed: Kamioka Observatory, Japan; Laboratori Nazionale del Gran Sasso (LNGS), Italy; and SNOLab, Canada. At each site, we compare the geo-$\bar{v}_e$ signal derived from these maps with the calculated background of $v_e$s from the Sun and $\bar{v}_e$s from nuclear reactors around the world. Assuming 10 live-years of operation, we find that 10-tonne-scale direction-sensitive detectors are capable of rejecting the solar-$v_e$ background and resolving the predicted $^{40}$K geo-$\bar{v}_e$ flux at 95% confidence level (CL). Similarly, 200-tonne-scale direction-sensitive detectors can identify the predicted mantle geo-$\bar{v}_e$ flux at 95% CL, while 20-kilotonne-scale detectors can probe radioactivity in the core.

## Results

**Neutrino flux model.** We consider the following neutrino fluxes: geo-$\bar{v}_e$s from the Earth's crust, mantle and core (Supplementary Note 1); solar $v_e$s from the Sun; and reactor $\bar{v}_e$s from worldwide nuclear power reactors. Interaction rates for atmospheric and relic supernova neutrinos are estimated to be negligible. A schematic representation illustrating these neutrino sources and their incident direction with respect to a terrestrial detector is shown in Supplementary Fig. 1.

**Geo-neutrinos.** The flux of geo-$\bar{v}_e$s at the surface of the Earth depends on the quantity and distribution of heat-producing elements, specifically U, Th and K, in the Earth's crust, mantle and core. The angular distribution of geo-$\bar{v}_e$s is modelled by calculating the geophysical response from eight potentially distinct geochemical reservoirs: sediments, upper, middle and lower continental crust, continental lithospheric mantle, oceanic crust, mantle and core. Seawater is not considered here, but estimated to contribute negligibly ($<0.05\%$) to the total geo-$\bar{v}_e$ flux. The geophysical response, expressed in g cm$^{-2}$, maps the spatial distribution and density variations of a given reservoir and is shown in Fig. 1 for each of the seven silicate (non-core) Earth reservoirs. The geo-$\bar{v}_e$ flux is then calculated from the geophysical response maps by assuming a specified abundance of U, Th and K in each reservoir (see Methods for details).

Table 1 lists the predicted fluxes of geo-$\bar{v}_e$s from U, Th and K decay in the Earth's crust, mantle and core, together with their respective uncertainties, at the three underground sites. We consider two sub-crustal Earth models, one with no radioactivity in the core and another with 10 p.p.b. U, Th in the core, both of which correspond to a total radiogenic heat of $18 \pm 11$ TW. The predicted incident angular distribution of geo-$\bar{v}_e$s from the Earth's crust and mantle at Gran Sasso is shown in Fig. 2a,b, respectively.

**Solar neutrinos.** Solar-$v_e$ flux predictions, presented in Supplementary Table 1, are taken from a recent global analysis[17] of solar and terrestrial neutrino data in the framework of three-neutrino mixing. The model predictions are all either in

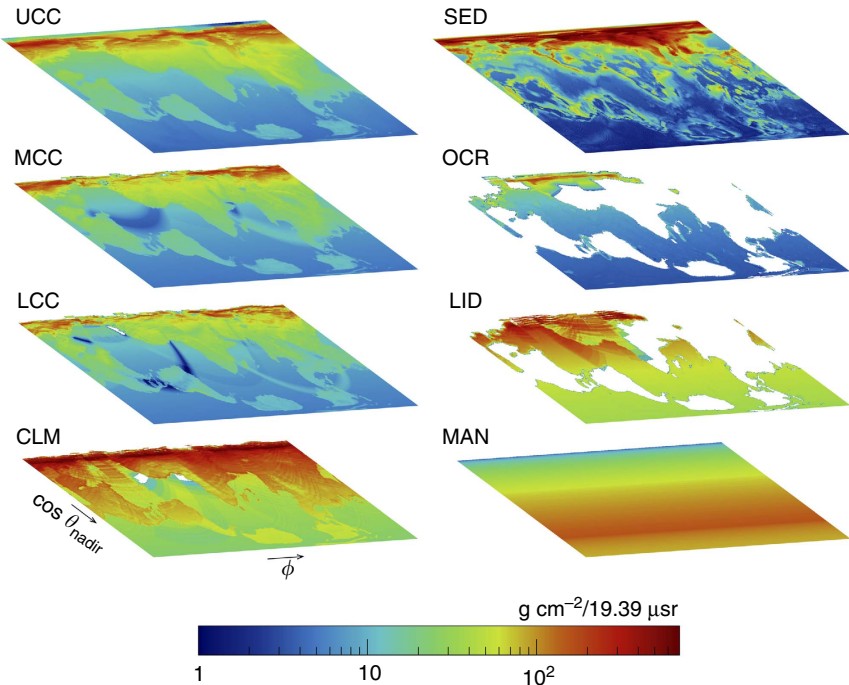

**Figure 1 | Geophysical response of seven silicate geochemical reservoirs.** Geophysical response (g cm$^{-2}$) calculated at Gran Sasso (Italy) and shown here for the upper continental crust (UCC), middle continental crust (MCC), lower continental crust (LCC), continental lithospheric mantle (CLM), sediments (SED), oceanic crust (OCR) and mantle (LID + MAN). The mantle is separated into a spherical shell (MAN) and a small volume (LID) sitting atop that breaks the spherical symmetry at a given location. The continental lithospheric mantle marks the transition between continental crust and mantle, about 80 km on average beneath the surface of the Earth. Values for SED (MAN) have been scaled up (down) by a factor of 10 (50) for the purposes of this figure. The azimuthal angle $\phi$ is measured counterclockwise from due North, while $\theta_{nadir}$ is measured with respect to the direction pointing to the centre of the Earth.

**Table 1 | Predicted flux of geo-neutrinos at three underground sites.**

| Site | Reservoir | $\Phi(^{238}U)$ ($10^6$ cm$^{-2}$ s$^{-1}$) | $\Phi(^{235}U)$ ($10^6$ cm$^{-2}$ s$^{-1}$) | $\Phi(^{232}Th)$ ($10^6$ cm$^{-2}$ s$^{-1}$) | $\Phi(^{40}K)$ ($10^6$ cm$^{-2}$ s$^{-1}$) |
|---|---|---|---|---|---|
| Kamioka (Japan) | Crust | 2.50 ± 0.65 | 0.08 ± 0.02 | 2.34 ± 0.35 | 10.0 ± 1.5 |
| | Mantle | 1.01 (0.71) ± 0.83 | 0.03 (0.02) ± 0.03 | 0.70 (0.63) ± 0.64 | 5.08 (3.56) ± 4.21 |
| | Core | 0.000 (0.300) | 0.000 (0.009) | 0.000 (0.066) | — |
| | Total | 3.51 ± 1.05 | 0.11 ± 0.03 | 3.04 ± 0.73 | 15.1 (13.6) ± 4.5 |
| Gran Sasso (Italy) | Crust | 3.18 ± 0.79 | 0.10 ± 0.02 | 3.00 ± 0.43 | 12.7 ± 1.8 |
| | Mantle | 1.00 (0.70) ± 0.82 | 0.03 (0.02) ± 0.03 | 0.70 (0.63) ± 0.63 | 5.05 (3.54) ± 4.18 |
| | Core | 0.000 (0.301) | 0.000 (0.009) | 0.000 (0.066) | — |
| | Total | 4.19 ± 1.14 | 0.13 ± 0.04 | 3.70 ± 0.77 | 17.8 (16.3) ± 4.6 |
| SNOLab (Canada) | Crust | 3.51 ± 0.91 | 0.11 ± 0.03 | 3.32 ± 0.48 | 14.4 ± 2.1 |
| | Mantle | 1.01 (0.71) ± 0.82 | 0.03 (0.02) ± 0.03 | 0.70 (0.63) ± 0.64 | 5.06 (3.54) ± 4.19 |
| | Core | 0.000 (0.301) | 0.000 (0.009) | 0.000 (0.066) | — |
| | Total | 4.52 ± 1.23 | 0.14 ± 0.04 | 4.02 ± 0.79 | 19.5 (18.0) ± 4.7 |

Geo-neutrino flux ($\Phi$) at Kamioka (Japan), Gran Sasso (Italy) and SNOLab (Canada) associated to uranium (U), thorium (Th) and potassium (K) decays in the crust, mantle and core, assuming a sub-crustal Earth model with a homogeneous mantle and 0 (10) p.p.b. U, Th in the core. Values given here do not include the effect of neutrino oscillation. Uncertainties are calculated by propagating the uncertainties on the elemental abundances from Supplementary Table 5.

good agreement with recent measurements[18–20] or reasonably well constrained[18,21]. Figure 2c shows the predicted incident angular distribution of solar $v_e$s at Gran Sasso.

**Reactor antineutrinos.** The intensity of $\bar{v}_e$s from nuclear power reactors is calculated at each underground site using positions and average powers of operating cores[22] from 2014 (or 2010), assuming a spherical Earth. Supplementary Table 2 lists the flux of reactor $\bar{v}_e$s at each underground site, along with the total

estimated uncertainty due to several potential sources of error (see Methods for details). Figure 2d shows the predicted incident angular distribution at Gran Sasso of $\bar{v}_e$s from worldwide nuclear power reactors.

Note that the fluxes presented in Table 1 and Supplementary Tables 1 and 2 do not include the effect of neutrino oscillation, which reduces the number of neutrinos by a factor[19] of $(1 - 0.5 \sin^2 2\theta_{12}) = 0.55$, on average, in the energy region of interest (Supplementary Note 2).

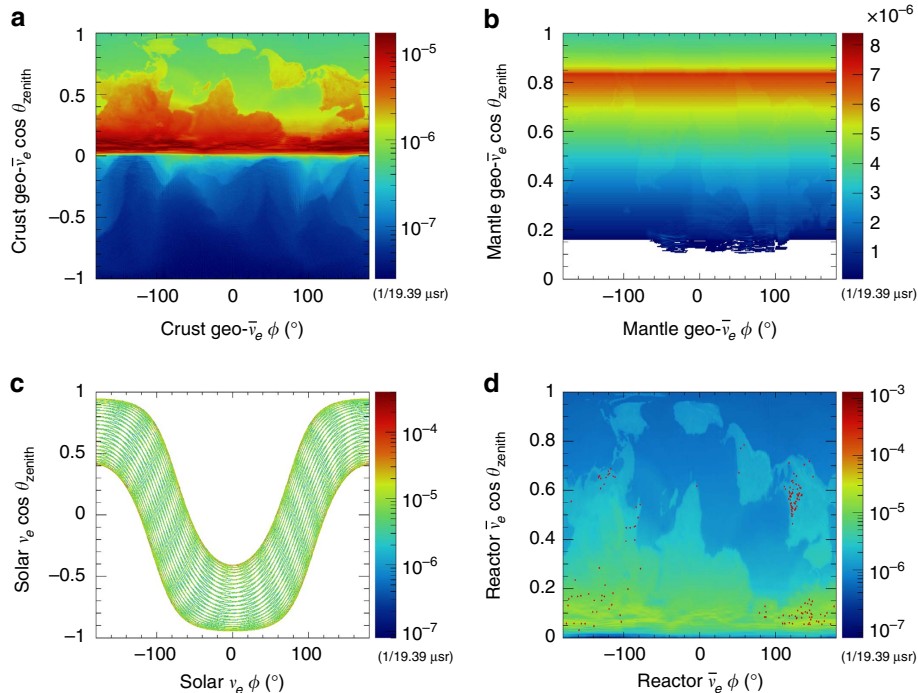

**Figure 2 | Angular distribution of incident neutrinos at Gran Sasso.** Angular distributions calculated at Gran Sasso (Italy) and shown here for incident neutrinos from: (**a**) the Earth's crust; (**b**) the Earth's mantle; (**c**) the Sun; and (**d**) worldwide nuclear power reactors. The azimuthal angle $\phi$ is measured clockwise from due North, while the zenith angle $\theta_{zenith}$ is measured with respect to the vertical axis, defined opposite the direction pointing to the centre of the Earth. (**a,b**) Crust and mantle geo-neutrino distributions are derived using CRUST 1.0 (ref. 15) and PREM[16], supplemented with detailed topological information for $\cos\theta_{zenith} < 0$. (**c**) The solar neutrino distribution is integrated over 1 calendar year at a latitude of 42.45°. (**d**) The directions to nuclear power reactors[22] are shown here in red, superimposed on the crust geo-neutrino map for visualization. The cluster at $\phi = 120°$, $0.5 < \cos\theta_{zenith} < 0.6$ is due to nuclear power reactors in the eastern half of the United States, while the cluster at $100° < \phi < 180°$, $0.05 < \cos\theta_{zenith} < 0.15$, and continuing to $-180° < \phi < -100°$, is due to nuclear power reactors in Western Europe, particularly France. Nuclear power reactors in South America, South Africa, Japan and India are also visible here. All plots are normalized to unit volume.

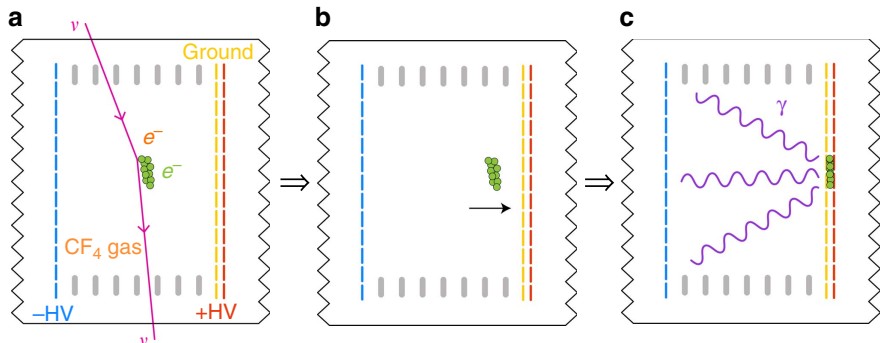

**Figure 3 | Schematic representation of the detection principle in a direction-sensitive detector.** A detector vessel is filled with $CF_4$ (or other) gas at low or high pressure. A uniform electric field is created by applying a potential difference across cathode (blue) and ground (yellow) planes that are electrically joined via step-down field cage rings (grey) at the outer extremity of the fiducial volume. (**a**) A target electron (orange) in the gas-filled volume recoils when struck by an incoming neutrino (magenta), depositing a substantial fraction of its kinetic energy as ionization (green) along its path. (**b**) The electric field then drifts this primary ionization to an amplification region, consisting of closely spaced ground and anode (red) planes with a strong electric field between them. (**c**) There, the ionization electrons undergo proportional multiplication, emitting electroluminescence or scintillation light (violet) and producing an image of the two-dimensional projection of the electron recoil track onto the amplification plane. If the amplification plane is read out with sufficient granularity and the diffusion in the drift volume is sufficiently small, the axial angle of the recoil track can be reconstructed, while the vector direction can be determined from the variation of the ionization density along the track. The track extent in the drift coordinate can additionally be reconstructed using fast readout of the charge time distribution of the avalanche signal.

**Direction-sensitive detectors.** Gas-filled time projection chambers (TPCs) fulfil the detection requirements posed by the geo-$\bar{v}_e$ measurements discussed here. Several groups[13,23,24] have developed TPCs to study physics requiring low backgrounds, a number of which[13,14,24,25] are capable of reconstructing the direction of low-energy recoils well below

the 1.3 MeV maximum energy of the $^{40}$K geo-$\bar{\nu}_e$ spectrum (Supplementary Note 3). The mechanism underlying these detectors is illustrated in Fig. 3.

In the following analysis, we assume detector resolution, efficiency (Supplementary Note 4), and energy thresholds based on the performance of previous detectors[13,14]. In a search for anomalous neutrino magnetic moment using $\bar{\nu}_e e^-$ elastic scattering, the MUNU collaboration[13] demonstrated electron energy and direction reconstruction performance in precisely the energy range relevant for the geo-$\bar{\nu}_e$ measurements proposed here. The detector was a 1 m$^3$ TPC, filled with CF$_4$ gas at 1 bar, instrumented with wire and strip readout at a pitch of 4.95 mm. The electron energy ($T$) resolution[13] was determined to be 10% at 200 keV and 6.8% at 478 keV, scaling as $T^{0.57}$. The measured electron angular resolution[13,26] varied from 15° at 200 keV to 12° at 400 keV and 10° at 600 keV. A best fit to these data points yields an energy dependence proportional to $T^{-0.361}$:

$$\sigma_\theta[°]=102\,T^{-0.361} \qquad (1)$$

with electron energy, $T$, in keV.

In addition to requirements on detector response and resolution, the geo-$\bar{\nu}_e$ measurements proposed here require detectors with large target masses and ultra-low background rates. High-pressure time projection chambers (HPTPCs), operating at pressures up to 10–15 bar, offer a promising solution. HPTPCs filled with gaseous xenon (Xe) are currently being used in searches for neutrino-less double beta decay[27,28] and have successfully demonstrated excellent energy resolution and background rates of $<4\times10^{-4}$ counts per keV per kg per year. Next-generation gaseous-Xe HPTPC experiments, such as PandaX-III[29], will feature target masses at the tonne scale.

For the geo-$\bar{\nu}_e$ sensitivity studies presented here, we assume that the techniques employed by experiments searching for dark matter[24,30,31] or neutrino-less double beta decay[27–29] can be used to mitigate backgrounds arising from the radioactivity of detector components, cosmogenic activation of the target gas, and any external neutron, photon and muon backgrounds. We consider the impact of these backgrounds on the estimated sensitivity in the Discussion section.

**Event rates**. We calculate the $\nu_\ell e^-$ elastic scattering cross-section for both $\nu_\ell$ and $\bar{\nu}_\ell$ following a standard example from the literature[32,33] (see Methods). Using the differential cross-section from equation (5), combined with the geo-$\bar{\nu}_e$, solar-$\nu_e$ and reactor-$\bar{\nu}_e$ flux values from Table 1 and Supplementary Tables 1 and 2, respectively, we calculate the predicted number of events per tonne-year exposure as a function of recoil electron kinetic energy, assuming a CF$_4$ target. Our calculation includes the effect of oscillation, $\nu_e \rightarrow \nu_\mu$, $\nu_\tau$ and subsequent $\nu_\mu e^-$ or $\nu_\tau e^-$ elastic scattering.

The predicted event rates, including the effect of oscillation, are shown in Fig. 4 and listed in Supplementary Tables 3 and 4. The geo-$\bar{\nu}_\ell$ signal to solar-$\nu_\ell$ background ratio is maximum for an electron energy threshold of 200 keV. In the following section, energy thresholds of 200, 250 and 800 keV are chosen to maximize the signal-to-background ratio and sensitivity of each study.

**Geo-neutrino sensitivity analysis**. We estimate the sensitivity of multi-tonne-scale direction-sensitive detectors to three sources of geo-$\bar{\nu}_e$s: $^{40}$K decays, the Earth's mantle and the Earth's core. Our methodology is described in the Methods section. Briefly, we simulate many pseudo-experiments and use a profile likelihood statistic to assess the exposure required to either set an upper limit for background-only pseudo-experiments, or exclude the

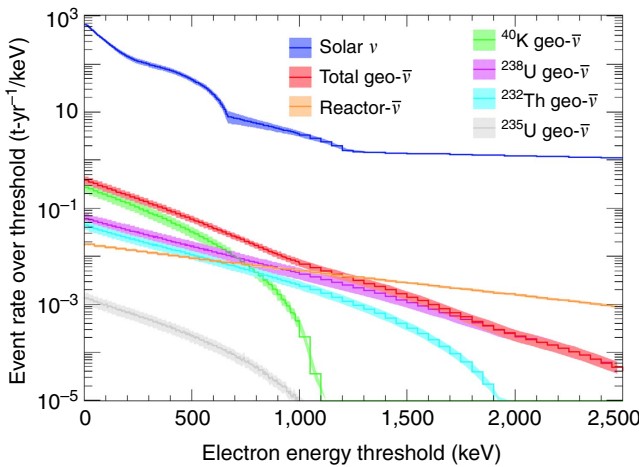

**Figure 4 | Predicted event rate over threshold per tonne-year exposure.** Event rates over threshold shown here versus electron energy threshold (keV) for solar neutrino (blue), geo-neutrino (red) and reactor antineutrino (orange) events, assuming a CF$_4$ target and 45% probability of oscillation into $\nu_\mu$ or $\nu_\tau$. Individual contributions to the total geo-neutrino flux from $^{40}$K (green), $^{238}$U (violet), $^{232}$Th (teal), and $^{235}$U (grey) decays are also shown. Geo-neutrino and reactor antineutrino rates are calculated using the normalizations at Gran Sasso from Table 1 and Supplementary Table 2, respectively. Error bands shown here correspond to uncertainties given in Supplementary Tables 3 and 4. Event rates are ∼0.4% higher for a SF$_6$ target, or 13.8% lower for a Xe target.

null hypothesis for signal + background pseudo-experiments at 95%, or 90%, CL.

For all studies, a flat $\pm7\%$ systematic uncertainty is applied on the pseudo-data, intended to cover measurement uncertainties on the global acceptance, track reconstruction efficiency, cross-section, number of targets in the fiducial volume, energy resolution and angular resolution, in addition to uncertainties on oscillation parameters. This value is consistent with the systematic uncertainty reported by the MUNU collaboration[26]. A $^{+1.5\%}_{-0.8\%}$ systematic uncertainty on the normalization of the total solar-$\nu_\ell$ flux is also applied (Supplementary Table 1). This results in a $^{+11.2\%}_{-5.3\%}$, $^{+12.3\%}_{-5.8\%}$ or $^{+20.0\%}_{-11.5\%}$ change in the solar-$\nu_\ell$ event rate for an energy threshold of 200, 250 or 800 keV, respectively (Supplementary Table 4). Similarly, a $\pm6\%$ uncertainty on the reactor-$\bar{\nu}_e$ flux and corresponding event rate is also applied to cover uncertainties on the energy spectrum, reactor powers and oscillation parameters (Methods, Supplementary Tables 2 and 3). The remaining systematic uncertainty on the geo-$\bar{\nu}_e$ background varies according to each study and is discussed in the respective sections below. Energy thresholds have been chosen to optimize the sensitivity of each analysis (that is, minimize the required exposure), assuming the angular resolution parameterization given in equation (1).

**$^{40}$K geo-neutrino flux**. We first estimate the sensitivity of a direction-sensitive detector to a non-zero $^{40}$K geo-$\bar{\nu}_e$ flux. For electron recoil energies below the 1.3 MeV endpoint energy of $^{40}$K $\beta^-$ decay, the detector proposed here would be unable to distinguish, on an event-by-event basis, the radioactive isotope from which an observed geo-$\bar{\nu}_\ell$-induced event was produced (Supplementary Note 5). A measurement of the $^{40}$K geo-$\bar{\nu}_e$ flux therefore benefits from prior experimental knowledge of the contributions from U and Th decays.

We use the most recently published measurements at Kamioka[4] and Gran Sasso[7] to constrain our model prediction

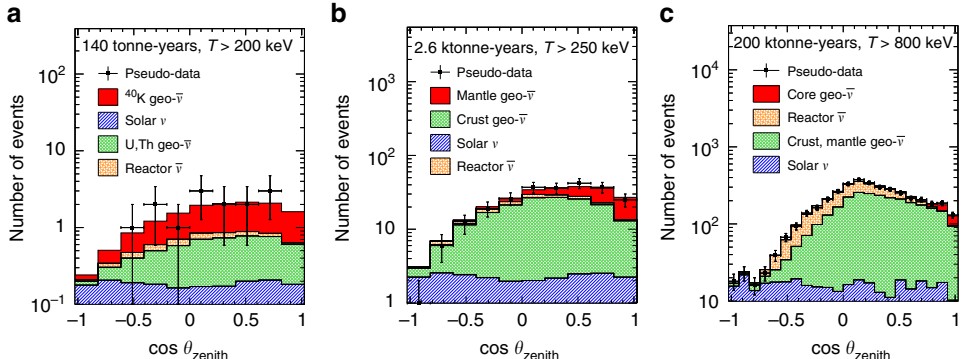

**Figure 5 | Zenith angle distribution of electron recoils induced by neutrinos.** Zenith angle distributions of electron recoils induced by signal geo-neutrinos from: (**a**) potassium ($^{40}$K) decays; (**b**) the mantle, assuming no radioactivity in the core; and (**c**) the core, assuming 10 p.p.b. uranium and thorium in the core. Signal-induced recoils (red solid) are shown together with recoils induced by background geo-neutrinos (green hatched), solar neutrinos (blue striped), and reactor antineutrinos (orange weave). Pseudo-experimental data (black squares) are also shown, with their statistical errors. Recoil distributions are calculated at Gran Sasso (Italy) and normalized to an exposure of (**a**) 140 tonne-years, (**b**) 2.6 ktonne-years or (**c**) 200 ktonne-years, including the effect of neutrino oscillation, with an applied electron energy threshold of $T >$ (**a**) 200 keV, (**b**) 250 keV or (**c**) 800 keV. Reconstructed angles have been smeared with a Gaussian distribution according to the angular resolution parameterization given in equation (1). The zenith angle ($\theta_{zenith}$) is measured with respect to the vertical axis, defined opposite the direction pointing to the centre of the Earth. A cut requiring a minimum angular separation from the Sun ($\theta_{sun}$) has been applied in order to reject the dominant background from solar neutrino events: $\cos \theta_{sun} <$ (**a**) $-0.09$, (**b**) 0.02 or (**c**) 0.54.

and estimate the background from U and Th geo-$\bar{\nu}_e$s (Table 1 and Supplementary Table 5). The uncertainties on the predicted geo-$\bar{\nu}_e$ flux result in a $\pm 20$, $\pm 18$ and $\pm 18\%$ change in the predicted geo-$\bar{\nu}_\ell$ background event rate at Kamioka, Gran Sasso and SNOLab, respectively, for an energy threshold of 200 keV (Supplementary Table 3). Although the uncertainties on these geo-$\bar{\nu}_e$ flux measurements may decrease between now and the completed construction of a multi-tonne-scale detector, we have instead opted for a conservative scenario using current measurement uncertainties.

To reject the background from solar-$\nu_\ell$ events, a cut on the angular separation from the Sun at the time of interaction, $\theta_{sun}$, is applied by fitting the measured spectrum (following scattering and angular smearing) for all recoil electrons with energy $T > 200$ keV with the functional form given in equation (9) (see Methods for details). The cut on electron energy has been optimized for this study to enhance the signal-to-background ratio (Fig. 4). Using the results of the fit, $\cos \theta_{sun}$ is required to lie outside of the range $\mu \pm 4\sigma$, or $\cos \theta_{sun} < -0.09$. Placement of the cut at $4\sigma$ has been optimized for this particular study and energy threshold. For an energy threshold of 200 keV and assuming the angular resolution parameterization given in equation (1), this cut accepts, on average, 44% of geo-$\bar{\nu}_\ell$ and reactor-$\bar{\nu}_\ell$ events (both signal and background), and 0.009% of solar-$\nu_\ell$ events. The acceptance was studied as a function of electron recoil energy, over the kinematic range $T < 2.0$ MeV, and seen to vary by at most 1.3% for geo-$\bar{\nu}_\ell$ events and reactor-$\bar{\nu}_\ell$ events, safely within the allotted $\pm 7\%$ measurement uncertainty.

For this study, we assume an Earth model with no radioactivity in the core. An example pseudo-experiment at Gran Sasso is shown in Fig. 5a. The results of the sensitivity analysis are presented in Table 2 and shown in Fig. 6a,b. The background-only measurement is statistics-limited for exposures $<200$ tonne-years, while the signal + background measurement is statistics-limited for exposures $< \sim 500$ tonne-years.

**Mantle geo-neutrino flux.** We next estimate the sensitivity of a direction-sensitive detector to the flux of geo-$\bar{\nu}_e$s coming from the Earth's mantle by treating all other sources, including geo-$\bar{\nu}_e$s from the Earth's crust, as background. From an optimization study, an

energy threshold of 250 keV is chosen not only to enhance the signal-to-background ratio (Fig. 4) but also remove lower energy tracks that scatter significantly and do not point along the incident neutrino direction. Similar to the previous study, a cut on $\cos \theta_{sun}$ is applied in order to remove the solar-$\nu_\ell$ background. For this study and energy threshold, $\cos \theta_{sun}$ is required to lie outside of the range $\mu \pm 4\sigma$, or $\cos \theta_{sun} < 0.02$. Assuming the angular resolution parameterization given in equation (1), this cut accepts, on average, 49% of crust and mantle geo-$\bar{\nu}_\ell$ events, 49% of reactor $\bar{\nu}_\ell$ events, and 0.008% of solar-$\nu_\ell$ events. Uncertainties on the predicted geo-$\bar{\nu}_e$ flux are also applied (Table 1), resulting in a $\pm 11\%$ change in the predicted background geo-$\bar{\nu}_\ell$ event rate at all sites (Supplementary Table 3).

For this study, we assume an Earth model with no radioactivity in the core. An example pseudo-experiment at Gran Sasso is shown in Fig. 5b. The results of the sensitivity analysis are presented in Table 2 and shown in Fig. 6c,d. The background-only measurement is statistics-limited for exposures $<1,500$ tonne-years. The signal + background measurement is statistics-limited for exposures $< \sim 3,000$ tonne-years.

**Core geo-neutrino flux.** Finally, we estimate the sensitivity of a direction-sensitive detector to the flux of geo-$\bar{\nu}_e$s coming from the Earth's core, assuming an abundance of 10 p.p.b. U and Th, but no K. All other sources, including geo-$\bar{\nu}_e$s from the Earth's crust and mantle, are treated as background. Based on optimization results, a higher energy threshold of 800 keV is used for this study in order to select a sample of electron recoils with good pointing resolution to the Earth's core, after the effects of scattering and angular smearing have been applied.

Similar to before, a cut on $\cos \theta_{sun}$ is applied in order to reject the solar-$\nu_\ell$ background. For this study and energy threshold, $\cos \theta_{sun}$ is required to lie outside of the range $\mu \pm 4.5\sigma$, or $\cos \theta_{sun} < 0.54$. Assuming the angular resolution parameterization given in equation (1), this cut accepts, on average, 78% of core geo-$\bar{\nu}_\ell$ events, 75% of crust geo-$\bar{\nu}_\ell$ events, 77% of mantle geo-$\bar{\nu}_\ell$ events, 74% of reactor $\bar{\nu}_\ell$ events and 0.03% of solar-$\nu_\ell$ events. A $\pm 7\%$ systematic uncertainty is applied to the geo-$\bar{\nu}_\ell$ background, assuming that the geo-$\bar{\nu}_e$ flux from $^{40}$K decays and the mantle have previously been measured, as described above.

**Table 2 | Mean geo-neutrino flux sensitivity at three underground sites.**

| $\Phi$ | Site (configuration) | Case (i) Background only (tonne-year) | Case (ii) Signal + background (tonne-year) | ($10^6$ cm$^{-2}$ s$^{-1}$) |
|---|---|---|---|---|
| $^{40}$K | Kamioka | 122 (85) | 157 (107) | 2.84 (3.28) |
| | Kamioka (2010) | 200 (130) | 279 (190) | 3.31 (3.67) |
| | Gran Sasso | 109 (75) | 140 (99) | 3.51 (4.01) |
| | Gran Sasso (high res.) | 70 (48) | 85 (58) | 3.38 (3.88) |
| | Gran Sasso (no syst.) | 106 (74) | 132 (94) | 3.63 (4.07) |
| | Gran Sasso (w/bkgnd) | 125 (86) | 161 (113) | 3.51 (4.12) |
| | SNOLab | 105 (72) | 132 (93) | 3.62 (4.18) |
| Mantle (no radioactivity in core) | Kamioka | 635 (437) | 1,980 (1,050) | 2.57 (2.51) |
| | Kamioka (2010) | 829 (567) | 2,630 (1,490) | 2.79 (2.80) |
| | Gran Sasso | 767 (523) | 2,560 (1,420) | 2.74 (2.73) |
| | Gran Sasso (high res.) | 525 (356) | 1,630 (933) | 2.71 (2.68) |
| | Gran Sasso (no syst.) | 705 (488) | 1,850 (1,050) | 2.50 (2.51) |
| | Gran Sasso (w/bkgnd) | 915 (608) | 3,010 (1,680) | 2.76 (2.77) |
| | SNOLab | 826 (564) | 2,680 (1,560) | 2.79 (2.83) |
| Core (10 p.p.b. U, Th in core) | Kamioka | 73,600 (53,300) | 215,000 (136,000) | 0.176 (0.185) |
| | Kamioka (2010) | 87,300 (59,000) | 343,000 (200,000) | 0.170 (0.176) |
| | Gran Sasso | 64,900 (47,600) | 202,000 (131,000) | 0.173 (0.183) |
| | Gran Sasso (high res.) | 54,400 (40,300) | 165,000 (102,000) | 0.172 (0.180) |
| | Gran Sasso (no syst.) | 62,000 (46,200) | 192,000 (126,000) | 0.180 (0.188) |
| | SNOLab | 64,000 (46,600) | 223,000 (131,000) | 0.171 (0.175) |

Mean sensitivity to the geo-neutrino flux ($\Phi$) from potassium ($^{40}$K) decays, the mantle, and the core, calculated at Kamioka (Japan), Gran Sasso (Italy) and SNOLab (Canada), assuming a CF$_4$ target and angular resolution given by equation (1). Case (i): exposure needed to achieve a mean 95% (90%) confidence level upper limit equal to the predicted flux, for background-only pseudo-experiments. Case (ii): exposure and mean lower limit at which 95% (90%) of signal + background pseudo-experiments have a non-zero 95% (90%) confidence level lower limit. An electron energy threshold of 200 keV ($^{40}$K), 250 keV (mantle) or 800 keV (core) has been applied. Systematic uncertainties for each study are discussed in the main text. Four variations of the baseline configuration are also presented here: (2010) uses reactor positions and powers from 2010, rather than 2014; (high res.) denotes improved angular resolution, equal to half that given by equation (1); (no syst.) denotes no systematic uncertainties; and (w/bkgnd) denotes inclusion of isotropic backgrounds, normalized to 20% ± 2% (62% ± 6%) of the $^{40}$K (mantle) geo-neutrino signal.

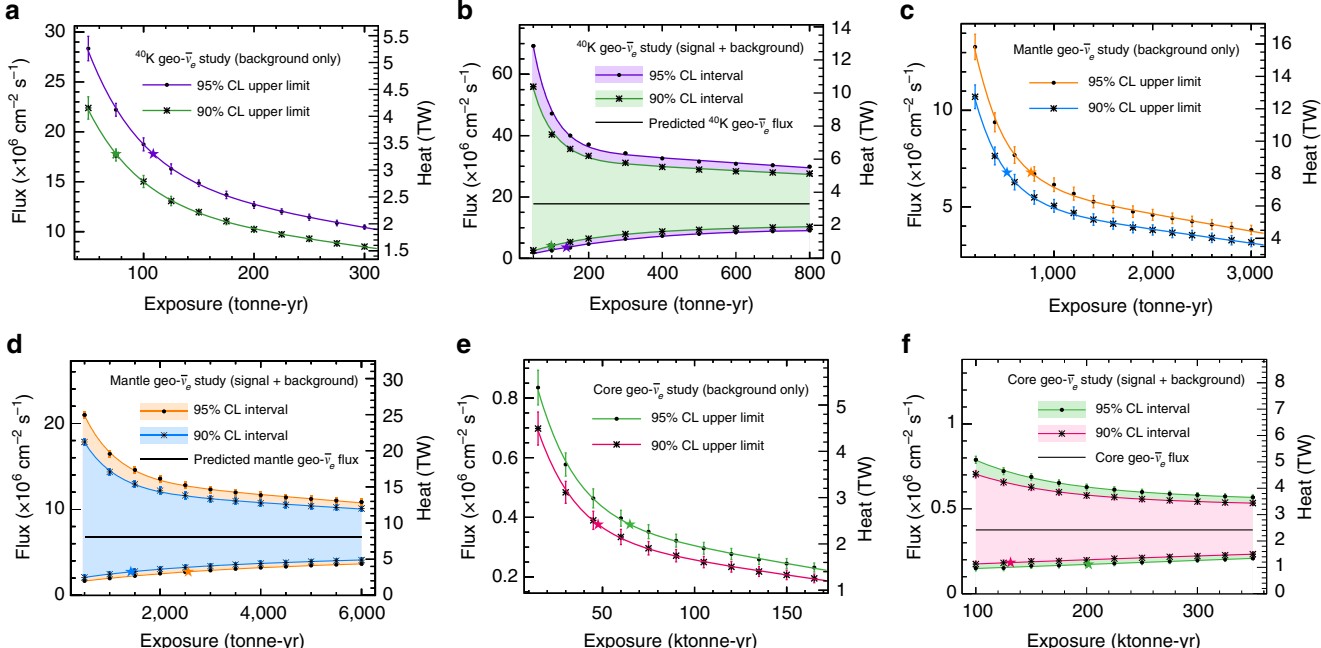

**Figure 6 | Mean geo-neutrino flux sensitivity at Gran Sasso.** Flux sensitivity calculated at Gran Sasso (Italy) and shown here versus exposure for the geo-neutrino flux from: (**a**,**b**) potassium ($^{40}$K) decays; (**c**,**d**) the mantle, assuming no radioactivity in the core; and (**e**,**f**) the core, assuming 10 p.p.b. uranium and thorium in the core. Error bars denote s.e.m. Each curve has been fit with a line plus exponential. (**a**,**e**) show the mean 95% (90%) confidence level (CL) upper limits for background-only pseudo-experiments; the star-shaped markers represent the exposures needed to achieve an upper limit equal to the predicted signal flux. (**b**,**d**,**f**) show the mean 95% (90%) confidence interval for signal + background pseudo-experiments; the star-shaped markers represent the exposures at which 95% (90%) of pseudo-experiments have a non-zero lower limit. A CF$_4$ target and angular resolution given by equation (1) have been assumed, with an applied electron energy threshold of $T >$ (**a**,**b**) 200 keV, (**c**,**d**) 250 keV or (**e**,**f**) 800 keV.

An example pseudo-experiment at Gran Sasso is shown in Fig. 5c. The results of the sensitivity analysis are presented in Table 2 and shown in Fig. 6e,f. The background-only measurement is statistics-limited for exposures < 90 ktonne-years. The signal + background measurement is statistics-limited for exposures $< \sim 180$ ktonne-years.

## Discussion

This paper introduces a method for measuring previously unresolved sources of radiogenic heating within the Earth associated to the geo-$\bar{v}_e$ flux from $^{40}$K, the mantle and the core. The technique exploits the directional information of $v_\ell e^-$ elastic scattering in order to positively identify rare signal interactions over significant backgrounds commonly thought of as irreducible. Gas-filled TPCs fulfil the detection requirement of tracking the direction of low-energy recoil electrons. The analysis presented here incorporates original, detailed mapping of geophysical structure, realistic assessments of neutrino background and the effects of detector response, in order to estimate the exposures needed to probe the geo-$\bar{v}_e$ flux from $^{40}$K, the mantle and the core. In each case, direction reconstruction, with modest angular resolution previously demonstrated by past experiments, mitigates the dominant background to the geo-$\bar{v}_e$ signal coming from solar $v_e$s.

Assuming ten live-years of operation and nuclear reactor positions and powers from 2014, we find that 10-tonne-scale direction-sensitive detectors have sensitivity to the predicted $^{40}$K geo-$\bar{v}_e$ flux at 95% CL at the three underground laboratories studied here: Kamioka (Japan), Gran Sasso (Italy) and SNOLab (Canada). With an exposure of 105–122 (72–85) tonne-years, a direction-sensitive detector with a CF$_4$ target and angular resolution parameterized by equation (1) can set a 95% (90%) CL upper limit on the predicted $^{40}$K geo-$\bar{v}_e$ flux of $15.1$–$19.5 \times 10^6\,cm^{-2}\,s^{-1}$ under the assumption that there is no signal. An exposure of 132–157 (93–107) tonne-years is needed to exclude the null hypothesis and set a non-zero 95% (90%) CL lower limit 95% (90%) of the time. For this study, the calculated exposures depend strongly on the angular resolution of the detector since it plays an important role in discriminating signal (geo-$\bar{v}_\ell$) versus background (solar-$v_\ell$) events, particularly for low-energy events that scatter considerably with respect to the incident neutrino direction. Enhancing the angular resolution of the detector by a factor of 2 results in a significant reduction in the required exposures: 39–41% reduction for the signal + background case and 36–37% reduction in the background-only case. In contrast, removing all systematic uncertainties reduces the required exposure only marginally, by 4.8–5.8% for the signal + background result and by 2.1–2.9% for the background-only result.

Under the same set of assumptions, we find that a 200-tonne-scale direction-sensitive detector has sensitivity to the predicted geo-$\bar{v}_e$ flux from the Earth's mantle at 95% CL, assuming no radioactivity in the core. With an exposure of 635–826 (437–564) tonne-years, a direction-sensitive detector can set a 95% (90%) CL upper limit on the predicted mantle geo-$\bar{v}_e$ flux of $6.8 \times 10^6\,cm^{-2}\,s^{-1}$ assuming that there is no signal, or set a non-zero 95% (90%) CL lower limit 95% (90%) of the time with an exposure of 1980–2680 (1050–1560) tonne-years. For this particular study, a detector at Kamioka requires the smallest exposure of the sites considered, due to Kamioka having the largest predicted ratio of mantle geo-$\bar{v}_e$s (signal) to crust geo-$\bar{v}_e$s (background). The higher predicted ratio of mantle-to-crust flux at Kamioka is a consequence of combining the assumption of a homogeneous mantle underneath all sites with the lower total geo-$\bar{v}_e$ flux measured by KamLAND, relative to Borexino. As

before, the results depend strongly on the assumed angular resolution since it is a crucial ingredient in the discrimination between signal geo-$\bar{v}_e$s from the mantle and background geo-$\bar{v}_e$s from the crust. Enhancing the angular resolution of the detector by a factor of 2 results in a 34–36% reduction in the required exposure for the signal + background case, or a 32% reduction for that of the background-only case. Systematic uncertainties play a larger role here than in the previous study, particularly in the signal + background case. Removing all considered uncertainties reduces the required exposure by 26–28% in the signal + background case and by 6.7–8.1% in the background-only case.

Finally, we find that 20-ktonne-scale direction-sensitive detectors have sensitivity to the predicted geo-$\bar{v}_e$ flux from the Earth's core at 95% CL, assuming a concentration of 10 p.p.b. of U and Th in the core. Although this target mass is perhaps beyond the feasibility of current gaseous detectors, the results are included here for completeness. A direction-sensitive detector can set a 95% (90%) CL upper limit on the predicted core geo-$\bar{v}_e$ flux of $0.38 \times 10^6\,cm^{-2}\,s^{-1}$ with an exposure of 64–74 (47–53) ktonne-years, assuming that there is no signal. To set a non-zero 95% (90%) CL lower limit 95% (90%) of the time, an exposure of 202–223 (131–136) ktonne-years is needed. Angular resolution plays a less critical role for this study since the higher energy threshold of 800 keV ensures smaller angular scattering of the electron recoil with respect to the incident neutrino. Improving the angular resolution by a factor of 2 reduces the exposures required by 18–23% for the signal + background case and by 15–16% for the background-only case. Removing all systematic uncertainties reduces the required exposure by 4.1–4.7% for the signal + background case and by 2.9–4.4% for the background-only case.

In the baseline configuration described above, nuclear reactor positions and powers from 2014 have been assumed, reflecting a scenario in which all Japanese reactors remain shutdown throughout operation and data-taking. The impact of this assumption on the results of these studies is estimated by instead using reactor positions and powers from 2010, reflecting an alternative scenario in which all Japanese reactors have resumed operation at the start of data-taking. The impact of this 2010 scenario is largest at the Kamioka site due to the proximity of Japanese reactors and is presented in Table 2. For the signal + background case, the calculated exposures increase by 77–83%, 33–42%, and 47–59% for the $^{40}$K, mantle, and core geo-$\bar{v}_e$ studies, respectively. For the background-only case, the required exposures increase by 52–64%, 30–31% and 11–19%, respectively. These results could likely be improved by imposing additional cuts on energy (since the reactor energy spectrum is harder than that of geo-$\bar{v}_e$s) or on the angular separation to the nearest reactor(s).

The exposures quoted above assume perfect rejection of non-neutrino backgrounds, namely those due to radioactive contamination of the detector or shield materials, intrinsic radioactivity or cosmogenic activation of the gas target, or other backgrounds coming from the cavern walls. To assess the impact of these non-neutrino backgrounds, we include an isotropic background at the level of 20 ± 2% (62 ± 6%) of the $^{40}$K (mantle) geo-$\bar{v}_e$ signal. This background level corresponds to a factor of $\sim 600$ lower than that achieved by the NEXT-100 experiment[27], after scaling to the energy region of interest (200 keV to 1 MeV) using the shape of the electronic recoil spectrum measured by the XENON100 collaboration[31] and removing the contribution from photomultiplier tubes, which account for $\sim 25\%$ of the background rate[27]. The presence of this background increases the required exposures by approximately 15% (16–19%) for the $^{40}$K (mantle) geo-$\bar{v}_e$ case.

While challenging, an improvement in background levels at the scale assumed here is plausible, given extrapolations from current experiments searching for dark matter and neutrino-less double

beta decay. Relative to the background rate projected by the NEXT-100 experiment, a substantial portion of this improvement would be attributed to the scale-up in fiducial volume, which reduces the relative background contribution coming from the vessel walls (due to the increase in the ratio of fiducial volume to surface area) and further suppresses the probability of single-scatter events in the fiducial volume. A scale-up of a similar size has been proposed by the LUX-ZEPLIN (LZ) experiment[30], relative to its predecessor[34], with a projected improvement in the electronic recoil background rate of a factor of ~1,000. Additional background reduction can be achieved by using lower-background material for the detector vessel, such as the high-purity copper used to construct the pressure vessel for the TREX-DM experiment[35], a high-pressure TPC searching for low-mass dark matter. The radioactive contamination of the TREX-DM copper pressure vessel due to $^{238}U$, $^{232}Th$, and $^{40}K$ is a factor of 8, 3 and 9 lower, respectively, than that of the LZ titanium cryostat vessel[30]. The background rate can be reduced further by surrounding the detector vessel with an active liquid scintillator veto layer to act as an anti-Compton detector, or optimizing the energy region of interest taking into account the complete background model.

Assuming a $CF_4$ (Xe) target and an operating pressure of 10 bar, a 15-tonne detector would have a fiducial volume of 403 (260) $m^3$, or ~7.4 (6.4) m on one side. Such a detector would be capable of making the first measurement of the $^{40}K$ geo-$\overline{v}_e$ flux at 95% CL with 10 live-years of data. Our results show that the most important experimental parameter to optimize is the angular resolution. The choice of target gas and operating pressure is therefore a trade-off between density and angular resolution, since higher density gives a greater event rate per unit volume, while also increasing the probability of multiple scattering along the track. To limit the effect of multiple scattering (diffusion) in the drift region of the TPC, the detector should be segmented into many TPC units, each with a drift length of ~1 m, housed within a single-pressure vessel. This design would also serve to suppress backgrounds coming from the vessel walls, via identification of coincident scattering interactions in multiple TPCs. To design the optimal detector, new measurements of angular resolution as a function of pressure in target gases of interest are needed.

In summary, we have proposed a method for measuring previously unresolved components of Earth's radiogenic heating using neutrino-electron elastic scattering in low-background direction-sensitive tracking detectors. According to the results of this study, the exposures needed to access $^{40}K$, mantle, and core geo-neutrinos at 95% CL are 140 tonne-years, 2.6 ktonne-years, and 200 ktonne-years, respectively. Realization of these exposures faces the challenge of constructing gas-filled TPCs with large target mass that are capable of stable operation for many years. Experiments employing high-pressure TPCs have demonstrated the possibility of combining large target masses with ultra-low background rates, suggesting feasibility.

The measurements proposed here chart a course for pioneering exploration of the veiled inner workings of the Earth, with the potential to unlock important clues about the radiogenic heating within our planet. Moreover, the required detectors, which are capable of reconstructing the direction of low-energy particles, open sensitivity to a wide range of topics and applications outside of geophysics, including searches for CP violation in the neutrino sector, astrophysical point sources, searches for dark matter, nuclear reactor monitoring and nuclear non-proliferation.

## Methods

**Detector sites.** The geographical coordinates and elevations used for each of the three underground sites considered here are: Kamioka Observatory, Japan

(36.427 °N, 137.300 °E), 358 m above sea level; Laboratori Nazionale del Gran Sasso (LNGS), Italy (42.450 °N, 13.567 °E), 936 m above sea level; and SNOLab, Canada (46.475 °N, −81.201 °E), 1,761 m below sea level.

**Geo-neutrino flux calculation.** Geophysical response maps for the near-surface reservoirs (that is, all reservoirs other than the continental lithospheric mantle, mantle and core) are constructed at a given location using the thicknesses and densities given by CRUST 1.0 (ref. 15). CRUST 1.0 provides a constant average elevation for each 1° × 1° unit of surface area. While this granularity is generally sufficient for the studies presented here, the angular distribution of geophysical response for the upper continental crust does not accurately represent observation sites located in mountainous terranes, such as Gran Sasso and Kamioka. In these cases, detailed topographic information is used to supplement CRUST 1.0. The mountainous regions surrounding the experimental site at Gran Sasso can be seen in Fig. 2a for $\cos\theta_{zenith} < 0$.

Calculation of the geophysical response of the mantle uses data from the Preliminary Reference Earth Model (PREM)[16], supplemented by data from CRUST 1.0. The main mantle volume is modelled as a spherical shell, as defined by PREM, extending outward from the core-mantle boundary ($R = 3,480$ km) to about 80 km, on average, beneath the surface. Shell density decreases radially outward, except for a slight increase over the outermost 140 km. Sitting atop the shell and beneath oceanic crust is a small mantle volume, which breaks the spherical symmetry of the geophysical response at the considered observation sites. The slight difference in elevations of each of the sites also contributes a small amount to this asymmetry.

The geophysical response of the mantle is calculated assuming that all element concentrations are homogeneously distributed. There is, however, experimental evidence of inhomogeneities in the deep mantle. Seismologists have resolved two large structures, called large low-shear-velocity provinces[36], off the west coast of Africa and near Tahiti in the South Pacific, with indication that they are thermally and compositionally different from the surrounding mantle. While these deep mantle structures could result in anisotropies[37] in the geo-$\overline{v}_e$ signal, particularly interesting for a direction-sensitive detector, their size and location are not currently agreed upon and are therefore not considered further in this paper.

The geo-$\overline{v}_e$ flux at a given point on the surface of the Earth is calculated by summing each of the geophysical response maps and multiplying by well-established values for neutrino luminosities per unit mass ($g^{-1} s^{-1}$), loosely constrained geochemical estimates of element abundances in each reservoir (Supplementary Table 5), and natural abundances of the isotopes (Supplementary Table 6). The estimated average abundances used here are generally consistent with other published abundances[38,39] within the same reservoir. Geo-$\overline{v}_e$ energy spectra are taken from Enomoto[40] (Supplementary Fig. 2).

**Reactor antineutrino flux calculation.** Positions and powers of nuclear reactors are taken from a reference worldwide reactor model[22]. Reactor powers are averaged over 12 calendar months of data from 2014 (or 2010). Intensities of each of the 439 (or 444) contributing reactor cores are calculated at the three underground sites assuming a spherical Earth. The total reactor-$\overline{v}_e$ flux is calculated by scaling the intensities by the average number of $\overline{v}_e$s released per unit energy, then summing over all reactor cores. Assuming a flux of 6 $\overline{v}_e$s and 205 MeV per fission, the average number of $\overline{v}_e$s released per unit energy[8] is $1.83 \times 10^{11} J^{-1}$.

The total estimated uncertainty on the reactor-$\overline{v}_e$ flux includes several potential sources of error: uncertainties in the power value reported by plant operators, oscillation parameters, conversion of reactor power to flux and seasonal changes in the reactor power output due to load requirements on the user side, refuelling or long-term shutdown. Assuming that the power output of all reactors is known exactly, then the uncertainty[22] in the flux above the inverse beta decay threshold of 1.8 MeV is ~2.2–2.5%, dominated mostly by the uncertainty on oscillation parameters. The added uncertainty due to power reporting is difficult to estimate reliably and should be studied in more detail.

While the shape of the reactor-$\overline{v}_e$ spectrum is well understood above the inverse beta decay threshold, it is less certain below this energy. Here we parameterize the spectral probability density function of reactor $\overline{v}_e$s using the functional form[8]:

$$N(E_{\overline{v}}) = \frac{6}{2.0523} e^{-(0.3125E_{\overline{v}} + 0.25)^2}, \tag{2}$$

where $E_{\overline{v}}$ is in MeV. This parameterization is close to that given in the literature[32]. A 5.5% uncertainty on the reactor-$\overline{v}_e$ energy spectrum is absorbed as a flat systematic uncertainty on the reactor-$\overline{v}_e$ flux, giving a total systematic uncertainty of 6%.

**Neutrino scattering cross-section.** Many experiments detect neutrinos using $v_e e^-$ elastic scattering[13,19,26,33]. This reaction is advantageous because it has no threshold, the maximum recoil electron kinetic energy can be of order MeV and the direction of the outgoing electron is well correlated with the direction of the incident neutrino (Supplementary Fig. 3). The theoretical uncertainties on the scattering cross-section are at the few-percent level; this process is therefore used to search for new physics, for example, anomalous neutrino magnetic moment[13]. The magnitude of the cross-section[33] for momentum transfer $q$ is roughly $\sigma(q) \sim 9.2 \times (q/10 \, MeV) \times 10^{-44} \, cm^2$.

Following Vogel and Engel[32], the cross-section for $v_\ell e^-$ elastic scattering is calculated in terms of the recoil electron kinetic energy, $T$, and scattering angle relative to the incident neutrino direction, $\chi$. The scattering angle depends on the incident neutrino energy $E_v$ as

$$\cos \chi = \frac{E_v + m_e}{E_v} \left[ \frac{T}{T + 2m_e} \right]^{1/2} \tag{3}$$

with a maximum recoil kinetic energy given by

$$T_{\max} = \frac{2E_v^2}{2E_v + m_e}, \tag{4}$$

where $m_e$ is the mass of the target electron. The differential cross-section is

$$\frac{d\sigma}{dT} = \frac{G_F^2 m_e}{2\pi} \left[ (g_V + x + g_A)^2 + (g_V + x - g_A)^2 \left( 1 - \frac{T}{E_v} \right)^2 + (g_A^2 - (g_V + x)^2) \frac{m_e T}{E_v^2} \right] + \frac{\pi \alpha^2 \mu_v^2}{m_e^2} \left( \frac{1}{T} - \frac{1}{E_v} \right), \tag{5}$$

where

$$g_V = \begin{cases} 2\sin^2 \theta_W + \frac{1}{2}, & \text{for } v_e \\ 2\sin^2 \theta_W - \frac{1}{2}, & \text{for } v_\mu, v_\tau \end{cases} \tag{6}$$

$$g_A = \begin{cases} +\frac{1}{2}, & \text{for } v_e \\ -\frac{1}{2}, & \text{for } v_\mu, v_\tau \end{cases} \tag{7}$$

$$x = \frac{\sqrt{2}\pi \alpha \langle r^2 \rangle}{3 G_F}, \tag{8}$$

$\alpha$ is the strong coupling constant, $G_F$ is the Fermi coupling constant, $\theta_W$ is the weak mixing angle, $\langle r^2 \rangle$ is the neutrino charge radius and $\mu_v$ is the neutrino magnetic moment. We set the latter two constants to zero in our calculation, consistent with experimental results[33,41,42] thus far. For $\bar{v}$, $g_A \to -g_A$ in equation (5). This interference between charged and neutral currents lowers the cross-section by a factor of $\sim 2.4$ at geo-$\bar{v}_e$ energies.

Supplementary Fig. 3 shows the differential $v_\ell e^-$ elastic scattering cross-section as a function of scattering angle, $\chi$, for three representative neutrino energies, $E_v = 0.5$, 1.5 and 5.0 MeV, integrated over the allowed kinematic range, $0 < T < T_{\max}$. Supplementary Fig. 4 shows the angular distribution of electron recoils induced by neutrinos from the Earth's crust, the Earth's mantle, the Sun and worldwide nuclear power reactors, calculated at Gran Sasso, with angular smearing according to equation (1) and electron recoil energy $T > 250$ keV.

**Sensitivity analysis method.** For each contribution to the total geo-$\bar{v}_e$ flux and for a given exposure, we run a set of 1,000 pseudo-experiments, treating a single contribution as signal and all other sources, including other geo-$\bar{v}_e$s, as background. Template distributions are constructed by simulating a large sample ($\geqslant 500$ M each) of geo-$\bar{v}_\ell$, solar-$v_\ell$ and reactor-$\bar{v}_\ell$ events, with event times distributed randomly over the year, according to the predicted incident angular distributions shown in Fig. 2, the energy spectra from Supplementary Fig. 2, Supplementary Table 1 and equation (2), and the scattering cross-section and kinematics presented in the previous section. The number of events in each pseudo-experiment is fluctuated according to the statistical error on the predicted event rates from Supplementary Tables 3 and 4 and scaled to the total exposure of the pseudo-experiment in tonne-years.

We assume a track reconstruction efficiency of 100% for electron recoil energies $T > 200$ keV. This is a reasonable assumption given the performance of past detectors (Supplementary Note 4), provided there are sufficient measurement points along the track. Reconstructed angles are smeared with a Gaussian distribution according to the angular resolution parameterization given in equation (1). This is perhaps an overly conservative scenario since the as-built detector would likely have finer point resolution in the readout plane, relative to that of the MUNU detector.

We define $\theta_{sun}$ as the angular separation between the electron recoil direction and the Sun-to-Earth vector at a given event time. This variable is a powerful discriminant between solar-$v_\ell$ and geo-$\bar{v}_\ell$ events since the former mostly point parallel to the Sun (cos $\theta_{sun} = 1$), while the latter are distributed evenly from $-1 \leqslant \cos \theta_{sun} \leqslant 1$. For each of the studies described here, a cut on cos $\theta_{sun}$ is applied in order to reject background events associated with solar $v_e$s. Supplementary Fig. 5 shows the simulated $\theta_{sun}$ distribution for two pseudo-experiments at Gran Sasso, scaled to an exposure of (a) 1 ktonne-year and (b) 10 ktonne-years, for electron recoils with energies $> 250$ keV and $> 800$ keV, respectively. Together, the two plots illustrate the energy dependence of the angular spread of recoils relative to the incident neutrino direction. The pseudo-experimental data in this figure has been fit with a von-Mises distribution, summed with a flat pedestal, to describe the solar-$v_\ell$ plus reactor- and geo-$\bar{v}_\ell$ background contributions, respectively:

$$f(\cos \theta) = A \frac{e^{\kappa \cos(\theta - \mu)}}{2\pi I_0(\kappa)} + C, \tag{9}$$

where $I_0(\kappa)$ is the modified Bessel function of order 0 and $A$, $C$, $\mu$ and $\kappa$ are parameters of the fit. The parameters $\mu$ and $1/\kappa$ are analogous to the mean ($\mu$) and variance ($\sigma^2$) of a Gaussian distribution. The mean $\chi^2/N_{dof}$ for 1,000 pseudo-experiments, each scaled to an exposure of 1 ktonne-year, is 8.83/6. The results of the fit at a given energy threshold are used to determine the precise cut value on $\theta_{sun}$, following an optimization of the calculated sensitivity. The cut values used and their calculated acceptance of signal and background contributions are discussed in the main text for each of the geo-$\bar{v}_e$ contributions under study.

For each set of pseudo-experiments, we assess two statistics: (i) the level of the signal geo-$\bar{v}_e$ flux that can be excluded at 95% (90%) CL, under the hypothesis that the data contain only background events; and (ii) the level of the signal geo-$\bar{v}_e$ flux that excludes the null hypothesis at 95% (90%) CL, assuming that the data contain both background and geo-$\bar{v}_e$ signal. Confidence intervals are calculated using a profile likelihood analysis, taking into account systematic uncertainties on the measured signal and predicted background distributions, discussed in the main text. To assess (i), we calculate the 95% (90%) confidence interval upper limit on the predicted geo-$\bar{v}_e$ signal, assuming that the data contain only background events, and determine the exposure needed to achieve an upper limit equal to the predicted signal flux. To assess (ii), we calculate the 95% (90%) confidence interval, assuming that the data contain both signal and background events, and determine the exposure at which 95% (90%) of pseudo-experiments have a non-zero lower limit. Results have been cross-checked with the frequentist CL (CL$_s$, CL$_b$) computation[43], with the profile likelihood method chosen here performing marginally better.

**Data availability.** The data that support the findings of this study are available within the article and its Supplementary Information files, or from the corresponding author upon reasonable request.

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

## Acknowledgements

J.M. thanks Mark Chen, Enectali Figueroa-Feliciano and Nikolai Tolich for critical discussions at various stages of the study. M.L. thanks Cécile Lapoire for helpful consistency checks of the analysis. M.L. acknowledges support from the Marie Skłodowska-Curie Fellowship program, under grant 665919 (EU, H2020-MSCA-COFUND-2014), Ministerio de Economia, Industria y Competitividad (MINECO), Agencia Estatal de Investigación (AEI) and Fondo Europeo de Desarrollo Regional (FEDER), under grants FPA2014-77347-C2-2 and SEV-2012-0234. S.D. appreciates past support from the Cooperative Studies of the Earth's Deep Interior (CSEDI) and the Cooperative Institute for Dynamic Earth Research (CIDER) programs funded by the US National Science Foundation. J.M. acknowledges support from the MIT Pappalardo Fellowship program, the US Department of Energy (contract number DE-FG02-05ER41360), the European Research Council Project ERC StG 279980, the UK Science & Technology Facilities Council (STFC) grant ST/K002570/1 and the Leverhulme Trust grant number ECF-20130496.

## Author contributions

All authors reviewed the manuscript and contributed to the design of the analysis. S.D. and J.M. conceived the study. S.D. developed the geo-$\bar{\nu}_e$ and reactor $\bar{\nu}_e$ models and angular distributions. M.L. developed the analysis and produced results. M.L. and J.M. prepared the manuscript.

## Additional information

**Competing interests:** The authors declare no competing financial interests.

**Publisher's note**: 

