## [Peer Review File · Nature Communications]

Reviewers' comments:

Reviewer #1 (Remarks to the Author):

This paper presents an analysis of the potential sensitivity of large gas time projection chambers to geo-neutrinos; with the main purposes of providing angular information and lower detection threshold by using neutrino-electron scattering. I consider the topic very important and the technique discussed quite interesting, with some caveats discussed below. The paper is well written and interdisciplinary; so it fits well the mission of the journal. While, of course, I have no way to try reproducing their results that derive from a complex analysis, the general methods are sound and the authors are well respected, so I have no reason to doubt the validity of the analysis. In summary I think this should be published if the authors consider the following comments (in two classes of importance).

Essential comments:

1) I am not convinced that the background discussion is fair. Basically the authors only discuss backgrounds due to other neutrinos (from the sun, nuclear reactors and parts of the earth that are not interesting for a certain (subset of the) measurement(s)). They essentially ignore radioactive background in the or around the detector and I think this greatly reduces the interest of their work. I do understand that estimating radioactive backgrounds in a generic way is very hard and maybe impossible, yet without a serious assessment of them one is left with the feeling that the power of the method is unclear. The authors mention that neutrinoless double-beta decay (bnbn) and dark matter (DM) detectors using gas TPCs have been considered and they refer to the backgrounds of these detectors, but I have two problems with this approach: a) these are basically all proposed detectors and as far as I know the backgrounds described have not been confirmed in large size detectors and b) even if they were, the situation here is quite different. In the case of DM detectors the interesting region is at a substantially lower energy, where self-shielding from the active gas is much more efficient. In the case of bnbn the signal is a sharp line, making active (ie from the analysis) background rejection more effective. I think the authors should at least point out these differences or, better, try providing an assessment of these backgrounds by simulation. This is not easy because it requires some engineering detail, like the thickness of the TPC vessel that, in turn, depends on the gas pressure.

2) Apart for the background issue (see 1 above) the main challenge here is the size of the detector and hence its practicality. In a sense, the paper may describe a "novel way" but it is not like no-one thought about this before. The issue is what is to be considered practical and what not. So more care should be given to demonstrating what size detectors we are talking about and why this is plausible. In line 202 they say 10 tonne to 10 ktonne -scale; that's a large range! And, anyway, what they really use are even larger sizes (although this is a bit confusing in my opinion, see 4 below). Taking Fig 7 as template there are 3 cases: 500 tonne-yr, 2000 tonne-yr and 200ktonne-yr. Now if I assume 10 yr data taking (that seems like a reasonable upper limit --and they should clearly state this, BTW) these correspond to 50tonne, 200tonne and 20ktonne detectors. Particularly for the smaller case --that is the more likely to be practical-- this is 50 instead of the original 10. In any case all these numbers have to be internally consistent. Also if I take the density they claim in lines 133-134 ($964\text{kg}/61\text{m}^3$), cubic detectors have a side of 14, 22 and 100 m for the 3 cases above. At least some of these seem really beyond practical. They hint that detectors can be segmented in smaller units (very many of them!) but then the background (from radioactivity that is typically in the vessel walls and outside) scales badly and without an assessment of it I really do not know what this all means. I also note that the position resolution that is used comes from that achieved at the MuNu experiment at Bugey (line 111). This had a $\sim 1\text{m}$ long drift and the authors should at least estimate the effects of diffusion for larger detectors.

Much less important comments:

3) In the reactor fluxes (table 3) they use to calculate this kind of background, Kamioka is analyzed with the current situation of most Japanese reactors off. I think they should also show the "pre-Tohoku earthquake" case as the paper discusses hypothetical measurements that will be done, if ever, many years from now. An analysis for both fluxes allows for bracketing what reality may be at the run time.

4) I think the organization of Figs 7 and 8 is not ideal and leaves the reader initially confused. Since the authors are really discussing 3 scales of detectors (some more practical than others, see 2)) why not reorganize Figs 7 and 8 into 3 figures, each consolidating the plots for one detector design.

5) (cosmetic) some of the curves of Fig 6 overlap in a way that make them hard to discern. This may improve a bit by limiting the vertical scale to the 10^{-5} - 10^4 range where most of the action is.

6) Line 324: "...likely due to Kamioka having..." Since the study is done by Monte Carlo it should be possible to extract what the real reason is, not the "likely" reason.

7) Fig 2 caption. "A vacuum chamber is evacuated..." sounds really weird and anyway it is not essential that the detector is a vacuum chamber. Possibly for very large detectors it would not be vacuum-proof and one may attain the appropriate gas purity by purge processes. How about "A detector vessel is filled..."

8) Fig 2: I would suggest indicating the flow of time between the 3 images from left to right.

Reviewer #2 (Remarks to the Author):

In this article, authors reported :

1. calculation result of angular distribution of (anti-)neutrinos
2. simulation result of sensitivity with future detectors

In terms of content 1, they could present the images of angular distribution (Figure 4 and Figure 5). These images help readers to understand that directional information provides now and unique opportunities to separate events in direction-sensitive detector. I would like to give comments on following things :

- They focused on LNGS to describe angular distributions. Is there any particular place to show the power of direction-sensitive detector (e.g. ocean, close to reactor)? That kind of discussion would give future prospect to readers.
- The file sizes of Figure 4 and Figure 5 are too large. Please reduce the data size.

In terms of content 2, it is clear that future large size TPC detectors can have sensitivity for 40K decay, mantle geo-neutrino and core geo-neutrino. The results are very impressive and motivate experiments to improve detection technic, however it is not so clear for readers that what kind of assumptions is conservative or optimistic. For example,

- background

In line 135-139, authors said that they neglected all sources of background other that solar neutrinos and reactor neutrinos. Could author roughly estimate background rate of existing TPC detectors in energy region of interest for geo-neutrino measurement?

- systematic uncertainty

From line 208, authors discussed about systematic uncertainties. Is that the conservative way to assume Borexino systematic uncertainties for TPC detector? Since these detectors have completely different detection technic, the source of systematic uncertainties should be different. Could authors make explanation more clear?

- detection technic

There are still many things to improve for future large TPC detectors. Could author give comments on what kind technic should be improved for enlarge the volume and improve pressure of TPC detector which has enough angular resolution for geo-neutrino measurement?

Other comments on content 2 are :

- Could author show energy spectrum assuming TPC detector? Existing liquid scintillator detector only have energy spectrum to measure geo-neutrinos. Energy spectrum can enhance speciality of direction-sensitive detector such as Figure 7 and Figure 11.

- I would like to recommend that table of sensitivity study results makes readers to understand more clear. Could author summarise "Discussion" in a table?

This article showed future prospect of geo-neutrino measurement with new technic. I think that the article has good achievements, but I recommend authors to reconsider above suggestions before the article will be accepted.

We thank the referees for carefully reviewing our manuscript and for providing useful comments and feedback. The revised manuscript includes discussion and treatment of intrinsic backgrounds and additional discussion on feasibility, all of which we consider significant improvements.

Reviewers' comments:

Reviewer #1 (Remarks to the Author):

This paper presents an analysis of the potential sensitivity of large gas time projection chambers to geo-neutrinos; with the main purposes of providing angular information and lower detection threshold by using neutrino-electron scattering. I consider the topic very important and the technique discussed quite interesting, with some caveats discussed below. The paper is well written and interdisciplinary; so it fits well the mission of the journal. While, of course, I have no way to try reproducing their results that derive from a complex analysis, the general methods are sound and the authors are well respected, so I have no reason to doubt the validity of the analysis. In summary I think this should be published if the authors consider the following comments (in two classes of importance).

Essential comments:

1) I am not convinced that the background discussion is fair. Basically the authors only discuss backgrounds due to other neutrinos (from the sun, nuclear reactors and parts of the earth that are not interesting for a certain (subset of the) measurement(s)). They essentially ignore radioactive background in the or around the detector and I think this greatly reduces the interest of their work. I do understand that estimating radioactive backgrounds in a generic way is very hard and maybe impossible, yet without a serious assessment of them one is left with the feeling that the power of the method is unclear. The authors mention that neutrinoless double-beta decay (bbnn) and dark matter (DM) detectors using gas TPCs have been considered and they refer to the backgrounds of these detectors, but I have two problems with this approach: a) these are basically all proposed detectors and as far as I know the backgrounds described have not been confirmed in large size detectors and b) even if they were, the situation here is quite different. In the case of DM detectors the interesting region is at a substantially lower energy, where self-shielding from the active gas is much more efficient. In the case of bbnn the signal is a sharp line, making active (ie from the analysis) background rejection more effective. I think the authors should at least point out these differences or, better, try providing an assessment of these backgrounds by simulation. This is not easy because it requires some engineering detail, like the thickness of the TPC vessel that, in turn, depends on the gas pressure.

>> Thank you for raising this important point. The aim of the paper is to explore the potential of using direction-sensitive detectors for neutrino geophysics, rather than to propose a specific detector design. To assess the impact of intrinsic backgrounds on the sensitivity study, we have re-run the K-40 and mantle analyses including an isotropic background and added the results of this variation to Table 6. We have also added two paragraphs to the Discussion section, including some design considerations and techniques for reducing these backgrounds, relative to the level measured by current experiments:

“The exposures quoted above assume perfect rejection of intrinsic backgrounds, namely those due to radioactive contamination of the detector and shield materials, intrinsic radioactivity or cosmogenic activation of the gas target, and any backgrounds coming from the cavern walls. To assess the impact of intrinsic backgrounds, we include an isotropic background at the level of 20% +/- 2% (62% +/- 6%) of the 40K (mantle) geo-antineutrino signal. This background level corresponds to a factor of ~600 lower than that achieved by the NEXT-100 experiment, after scaling to the energy region of interest relevant for this analysis (200 keV to 1 MeV) using the shape of the electronic recoil spectrum measured by the XENON100 collaboration and removing the contribution from photomultiplier tubes (PMTs). The presence of this background increases the required exposures by approximately 15% (16--19%) for the 40K (mantle) geo-antineutrino case.

While challenging, an improvement in background levels at the scale assumed here is plausible, given extrapolations from current experiments searching for dark matter and neutrino-less double beta decay. Relative to the background rate projected by the NEXT-100 experiment, a significant portion of this improvement would be attributed to the scale-up in fiducial volume, which reduces the relative contribution coming from the vessel walls (due to the increase in the ratio of fiducial volume to surface area) and further suppresses the probability of single-scatter events in the fiducial volume. A scale-up of a similar size has been proposed by the LUX-ZEPLIN (LZ) experiment, relative to LUX, with a projected improvement in the electronic recoil background rate of a factor of ~1000. Additional background reduction can be achieved by using lower-background material for the detector vessel, such as the high-purity copper used to construct the pressure vessel for the TREX-DM experiment, a high-pressure TPC searching for low-mass dark matter. The radioactive contamination due to ^{238}U , ^{232}Th , and ^{40}K of the TREX-DM copper pressure vessel is a factor of 8, 3, and 9 lower, respectively, than that of the LZ titanium cryostat vessel. The background rate can be reduced further by: surrounding the detector vessel with an active liquid scintillator veto layer to act as an anti-Compton detector; optimizing the energy region of interest taking into account the complete background model; and avoiding the use of PMTs in the active volume. PMTs account for approximately 25% of the background rate for NEXT-100.”

2) Apart for the background issue (see 1 above) the main challenge here is the size of the detector and hence its practicality. In a sense, the paper may describe a "novel way" but it is not like no-one thought about this before. The issue is what is to be considered practical and what not. So more care should be given to demonstrating what size detectors we are talking about and why this is plausible. In line 202 they say 10 tonne to 10 ktonne -scale; that's a large range!

>> The large range quoted in line 202, as well as other places in the text, was intended to cover the entire range of target masses needed to gain sensitivity to K40, mantle and core geo-neutrinos. The text has been edited in all instances, either by breaking the cases down into separate sentences, generalizing to “multi-tonne-scale”, or removing the “X-tonne-scale” modifier. Example:

“Assuming ten live-years of operation, we find that 10-tonne-scale direction-sensitive detectors are capable of rejecting the solar ν_e background and gaining sensitivity to the predicted 40K geo-antineutrino flux at 95% confidence level (CL). Similarly, direction-sensitive detectors at the 200-tonne (20-ktonne) scale are capable of gaining sensitivity to the predicted mantle (core) geo-antineutrino flux at 95% CL.”

And, anyway, what they really use are even larger sizes (although this is a bit confusing in my opinion, see 4 below). Taking Fig 7 as template there are 3 cases: 500 tonne-yr, 2000 tonne-yr and 200ktonne-yr. Now if I assume 10 yr data taking (that seems like a reasonable upper limit --and they should clearly state this, BTW) these correspond to 50tonne, 200tonne and 20ktonne detectors. Particularly for the smaller case --that is the more likely to be practical-- this is 50 instead of the original 10. In any case all these numbers have to be internally consistent.

>> The exposures used in Figure 7 were chosen so that readers could scale the y-axis with some easy mental math. However, we understand the confusion that this has caused and have updated the figures with the exposures required to resolve the signal with 95% CL: 140 tonne-yr, 2.6 ktonne-yr, and 200 ktonne-yr. We have also added the clause “Assuming ten live-years of operation” in various places to make this point clear to the reader (see quoted example directly above).

Also if I take the density they claim in lines 133-134 ($964\text{kg}/61\text{m}^3$), cubic detectors have a side of 14, 22 and 100 m for the 3 cases above. At least some of these seem really beyond practical. They hint that

detectors can be segmented in smaller units (very many of them!) but then the background (from radioactivity that is typically in the vessel walls and outside) scales badly and without an assessment of it I really do not know what this all means. I also note that the position resolution that is used comes from that achieved at the MuNu experiment at Bugey (line 111). This had a ~1m long drift and the authors should at least estimate the effects of diffusion for larger detectors.

>> We agree that a 20-ktonne detector is beyond the feasibility of current gaseous detectors and have added the following sentence to the Discussion section:

“This [20-ktonne] target mass is perhaps beyond the feasibility of current gaseous detectors, but the results are included here for completeness.”

We have added a paragraph to the Discussion section to address your concerns about feasibility:

“Assuming a CF₄ (Xe) target and an operating pressure of 10 bar, a 15-tonne detector would have a fiducial volume of 403 (260) m³, or approximately 7.4 (6.4) m on one side. Such a detector would be capable of making the first measurement of the 40K geo-antineutrino flux at 95% CL with ten live-years of data. Our results show that the most important experimental parameter to optimize is the angular resolution. The choice of target gas and operating pressure is therefore a trade-off between density and angular resolution, since higher density produces a greater event rate per unit volume, but also increases the probability of multiple scattering along the track. To limit the effect of multiple scattering (diffusion) in the drift region of the TPC, the detector should be segmented into many TPC units, each with a drift length of ~1 m, housed within a single pressure vessel. This segmentation would also serve to suppress backgrounds coming from the vessel walls, via identification of coincident scattering interactions in multiple TPCs^{}. To design the optimal detector, new measurements of angular resolution as a function of pressure in target gases of interest are needed.**

**** The interaction length of mid-energy (200 keV to 1 MeV) photons in 10 bar of CF₄ (Xe) is approximately 2.3--4.4 (0.5--3.0) m.”**

Much less important comments:

3) In the reactor fluxes (table 3) they use to calculate this kind of background, Kamioka is analyzed with the current situation of most Japanese reactors off. I think they should also show the "pre-Tohoku earthquake" case as the paper discusses hypothetical measurements that will be done, if ever, many years from now. An analysis for both fluxes allows for bracketing what reality may be at the run time.

>> We have re-run the Kamioka analysis using reactor positions and powers from 2010 and added the results of this variation to Table 6. The following passage has been added to the main text:

“Due to the shut down of all reactors in Japan following the Fukushima-Daichi incident on March 11, 2011, an additional variation specific to the Kamioka site is studied using reactor positions and powers from 2010. This case is intended to reflect a future scenario in which Japanese nuclear reactors have resumed operation.”

The following paragraph has been added to the Discussion section:

“In the baseline configuration, reactor positions and powers from 2014 have been assumed, reflecting a scenario in which all Japanese reactors remain shut down throughout operation and data-taking. The impact of this assumption on the results of the sensitivity analysis is quantified by instead using reactor positions and powers from 2010, reflecting a scenario in which all Japanese

reactors have resumed operation at the start of data-taking. The impact of this '2010' scenario is largest at the Kamioka site due to the proximity of Japanese reactors. The results are presented in Table 6. For the signal + background case, the calculated exposures increase by 77--83%, 33--42%, and 47--59% for the 40K, mantle, and core studies, respectively. For the background-only case, the required exposures increase by 52--64%, 30--31%, and 11--19%, respectively. These results could likely be improved by imposing additional cuts on energy (since the reactor spectrum is harder than that of geo-antineutrinos) or on the angular separation to the nearest reactor(s)."

4) I think the organization of Figs 7 and 8 is not ideal and leaves the reader initially confused. Since the authors are really discussing 3 scales of detectors (some more practical than others, see 2)) why not reorganize Figs 7 and 8 into 3 figures, each consolidating the plots for one detector design.

>> **We agree that this change might clear up some confusion, but we prefer to address this at a later stage in the editorial process since many of the figures (such as Fig. 7) will likely be moved to the Supplementary Materials section. Figure 8, however, summarizes the results of the analysis and will likely be kept in the main section. No change has been made.**

5) (cosmetic) some of the curves of Fig 6 overlap in a way that make them hard to discern. This may improve a bit by limiting the vertical scale to the 10^{-5} - 10^4 range where most of the action is.

>> **The figure has been updated. The vertical scale has been reduced to 10^{-5} to 10^3 and the legend has been moved to the side.**

6) Line 324: "...likely due to Kamioka having..." Since the study is done by Monte Carlo it should be possible to extract what the real reason is, not the "likely" reason.

>> **"likely" has been removed.**

7) Fig 2 caption. "A vacuum chamber is evacuated..." sounds really weird and anyway it is not essential that the detector is a vacuum chamber. Possibly for very large detectors it would not be vacuum-proof and one may attain the appropriate gas purity by purge processes. How about "A detector vessel is filled..."

>> **Edited accordingly.**

8) Fig 2: I would suggest indicating the flow of time between the 3 images from left to right.

>> **Arrows between panels have been added and 'left', 'middle' and 'right' descriptors have been added to the caption.**

Reviewer #2 (Remarks to the Author):

In this article, authors reported :

1. calculation result of angular distribution of (anti-)neutrinos
2. simulation result of sensitivity with future detectors

In terms of content 1, they could present the images of angular distribution (Figure 4 and Figure 5). These images help readers to understand that directional information provides new and unique opportunities to separate events in direction-sensitive detector. I would like to give comments on following things :

- They focused on LNGS to describe angular distributions. Is there any particular place to show the power of direction-sensitive detector (e.g. ocean, close to reactor)? That kind of discussion would give future prospect to readers.

>> We agree that there are many interesting topics to explore here, but we need to maintain brevity and focus since the manuscript is already long as is. Nuclear reactor monitoring will be covered in a separate manuscript. Ocean deployment of a direction-sensitive detector is certainly interesting, particularly for measuring anisotropies in the mantle, but a proper discussion is lengthy and requires supporting figures. We are considering this topic for a separate manuscript as well.

- The file sizes of Figure 4 and Figure 5 are too large. Please reduce the data size.

>> Figures have been replaced with lower-resolution versions.

In terms of content 2, it is clear that future large size TPC detectors can have sensitivity for 40K decay, mantle geo-neutrino and core geo-neutrino. The results are very impressive and motivate experiments to improve detection technic, however it is not so clear for readers that what kind of assumptions is conservative or optimistic. For example,

- background

In line 135-139, authors said that they neglected all sources of background other than solar neutrinos and reactor neutrinos. Could author roughly estimate background rate of existing TPC detectors in energy region of interest for geo-neutrino measurement?

>> Thank you for raising this important issue. Please see our response to point 1) above.

- systematic uncertainty

From line 208, authors discussed about systematic uncertainties. Is that the conservative way to assume Borexino systematic uncertainties for TPC detector? Since these detectors have completely different detection technic, the source of systematic uncertainties should be different. Could authors make explanation more clear?

>> The systematic uncertainty on pseudo-data has been changed to 7%, consistent with the value reported by the MUNU collaboration in their 2005 paper. We have re-run the entire analysis and updated the text and results:

“For all studies, a flat 7% systematic uncertainty is applied on the pseudo-data, intended to cover measurement uncertainties on the global acceptance, track reconstruction efficiency, cross section, number of targets in the fiducial volume, energy resolution and angular resolution, in addition to uncertainties on oscillation parameters. This value is consistent with the systematic uncertainty reported by the MUNU collaboration.”

- detection technic

There are still many things to improve for future large TPC detectors. Could author give comments on what kind technic should be improved for enlarge the volume and improve pressure of TPC detector which has enough angular resolution for geo-neutrino measurement?

>> The following paragraph has been added to the Discussion section, outlining some TPC design considerations and how best to scale up in fiducial volume. Additional measurements of angular resolution as a function of pressure are needed.

“Assuming a CF₄ (Xe) target and an operating pressure of 10 bar, a 15-tonne detector would have a fiducial volume of 403 (260) m³, or approximately 7.4 (6.4) m on one side. Such a detector would be capable of making the first measurement of the 40K geo-antineutrino flux at 95% CL with ten live-

years of data. Our results show that the most important experimental parameter to optimize is the angular resolution. The choice of target gas and operating pressure is therefore a trade-off between density and angular resolution, since higher density produces a greater event rate per unit volume, but also increases the probability of multiple scattering along the track. To limit the effect of multiple scattering (diffusion) in the drift region of the TPC, the detector should be segmented into many TPC units, each with a drift length of ~ 1 m, housed within a single pressure vessel. This segmentation would also serve to suppress backgrounds coming from the vessel walls, via identification of coincident scattering interactions in multiple TPCs**. To design the optimal detector, new measurements of angular resolution as a function of pressure in target gases of interest are needed.

** The interaction length of mid-energy (200 keV to 1 MeV) photons in 10 bar of CF₄ (Xe) is approximately 2.3--4.4 (0.5--3.0) m."

Other comments on content 2 are :

- Could author show energy spectrum assuming TPC detector? Existing liquid scintillator detector only have energy spectrum to measure geo-neutrinos. Energy spectrum can enhance speciality of direction-sensitive detector such as Figure 7 and Figure 11.

>> The energy spectrum has not been included in the manuscript since it is not essential for understanding the analysis and we are aiming to reduce the number of figures. An example from a pseudo-experiment normalized to 500 tonne-yr is included here:

- I would like to recommend that table of sensitivity study results makes readers to understand more clear. Could author summarise "Discussion" in a table?

>> All results mentioned in the Discussion section are also presented in Table 6. Is this satisfactory?

This article showed future prospect of geo-neutrino measurement with new technic. I think that the article has good achievements, but I recommend authors to reconsider above suggestions before the article will be accepted.

REVIEWERS' COMMENTS:

Reviewer #1 (Remarks to the Author):

Given the changes by the authors I have no basic objections to the publication of the paper. The paper is very long and in some of their comments the authors seem to imply that they do not expect it to be published as is but some of the material will be moved to a "supplementary section". I assume that the editor will follow this process (ie I can't judge it).

Reviewer #2 (Remarks to the Author):

The revised version of this manuscript has much more clear messages than first version. The discussion on background estimation is what I would like to know more about, and I do not have any unclear points about assumption of systematic uncertainty.

Thank you for providing the information on energy spectrum. I agree with the authors that there is no essential motivation to understand their way of analysis. However, it is interesting for me.

I think that this manuscript shows one of the directions for future prospect of geo-neutrino measurements. I would like to recommend that this manuscript should be accepted.

We kindly thank you for reviewing our manuscript and providing useful comments and feedback. We are especially grateful for expressing your support in regards to publication of the manuscript.

REVIEWERS' COMMENTS:

Reviewer #1 (Remarks to the Author):

Given the changes by the authors I have no basic objections to the publication of the paper. The paper is very long and in some of their comments the authors seem to imply that they do not expect it to be published as is but some of the material will be moved to a "supplementary section". I assume that the editor will follow this process (ie I can't judge it).

>> We have shortened the introduction considerably and moved some technical points, figures, and tables to a supplementary information file. For coherence, we decided to keep the three pseudo-experiments together in one figure, and the sensitivity results vs. exposure in another. However, we have modified the caption to carefully explain the difference between the different sub-panels, e.g.

“Zenith angle distributions of electron recoils induced by signal geo-neutrinos from: (a) potassium (^{40}K) decays; (b) the mantle, assuming no radioactivity in the core; and (c) the core, assuming 10 p.p.b. uranium and thorium in the core. Signal-induced recoils (red solid) are shown together with recoils induced by background geo-neutrinos (green hatched), solar neutrinos (blue striped), and reactor anti-neutrinos (orange weave). Pseudo-experimental data (black squares) are also shown, with their statistical errors. Recoil distributions are calculated at Gran Sasso (Italy) and normalized to an exposure of (a) 140 tonne-years, (b) 2.6 ktonne-years, or (c) 200 ktonne-years, including the effect of neutrino oscillation, with an applied electron energy threshold of $T >$ (a) 200 keV, (b) 250 keV, or (c) 800 keV.”

Reviewer #2 (Remarks to the Author):

The revised version of this manuscript has much more clear messages than first version. The discussion on background estimation is what I would like to know more about, and I do not have any unclear points about assumption of systematic uncertainty.

Thank you for providing the information on energy spectrum. I agree with the authors that there is no essential motivation to understand their way of analysis. However, it is interesting for me.

I think that this manuscript shows one of the directions for future prospect of geo-neutrino measurements. I would like to recommend that this manuscript should be accepted.

>> Thank you.